Potentially toxic metal(loid) distribution and migration in the bottom weathering profile of indigenous zinc smelting slag pile in clastic rock region

Peng Yishu 1
Yang Ruidong 2 rdyang@gzu.edu.cn
Jin Tao 3
Chen Jun 4
Zhang Jian 5
1 College of Tea Science, Guizhou University , Guiyang , China
2 College of Resources and Environmental Engineering, Guizhou University , Guiyang , China
3 Institute of Mountain Resources of Guizhou Province, Guizhou Academy of Sciences , Guiyang , China
4 State Key Laboratory of Ore Deposit Geochemistry, Institute of Geochemistry, Chinese Academy of Sciences , Guiyang , China
5 College of Environmental Science and Engineering, Yangzhou University , Yangzhou , China
Zuniga-Gonzalez Carlos
Electronic publication date: 2021 Apr 7
Publication date: 2021
Volume: 9
Electronic Location ID: e10825
Received 2020 Sep 16; Accepted 2021 Jan 3
Copyright: © 2021 Peng et al.
Copyright year: 2021
Copyright holder: Peng et al.
License: This is an open access article distributed under the terms of the Creative Commons Attribution License, which permits unrestricted use, distribution, reproduction and adaptation in any medium and for any purpose provided that it is properly attributed. For attribution, the original author(s), title, publication source (PeerJ) and either DOI or URL of the article must be cited.
License URL: https://creativecommons.org/licenses/by/4.0/

Keywords: Indigenous zinc smelting slag, Potentially toxic metal(loid), Migration, Weathering profile, Clastic rock region

Funding: National Natural Science Foundation of China 41463009 Guizhou Provincial Science and Technology Foundation QKHJC-ZK[2021]YB232 Public and Basic Geological Project of Guizhou Province QGTZF-2015-20 Scientific Research Project for Introducing Talents into Guizhou University GDRJHZ[2019]05 Foundation for Innovative Major Research Groups of the Education Bureau in Guizhou Province QJH-KY-2016-024 First-Class Subjects “Ecology” in Guizhou Province GNYL[2017]007 This project was financially supported by the National Natural Science Foundation of China (Grant No. 41463009), the Guizhou Provincial Science and Technology Foundation (QKHJC-ZK[2021]YB232), the Public and Basic Geological Project of Guizhou Province (No. QGTZF-2015-20), the Scientific Research Project for Introducing Talents into Guizhou University (GDRJHZ[2019]05), the Foundation for Innovative Major Research Groups of the Education Bureau in Guizhou Province (QJH-KY-2016-024) and the Construction Project of the First-Class Subjects “Ecology” in Guizhou Province (GNYL[2017]007). The funders had no role in study design, data collection and analysis, decision to publish, or preparation of the manuscript.

==============================
Background

There are contaminated by potentially toxic metal(loid)s (PTMs) that the surface soil and the weathering profiles around the indigenous zinc smelting slag piles or smelters in the smelting area. However, few systematic studies are currently focusing on the PTM distribution and migration among the slag and its bottom weathering profile.

Methods

This research determined the concentrations of PTMs and pH values. And we analyzed PTM distribution in the two weathering profiles (slag-covered and slag-absent) with a small horizontal distance in the clastic rock region in the smelting area.

Results

The soil As and Pb contents, respectively, within the 30 and 50 cm depth in the slag-covered section were higher than those in the slag-absent profile. All soil Cd and Zn contents of the slag-covered core were significantly higher than those in the slag-absent weathering section.

Conclusions

Compared with the slag-absent weathering section, some PTMs (i.e., As, Cd, Pb and Zn) in the bottom weathering profile were polluted by these elements in the covered slag in the clastic rock region, and their depths were influenced by the slag to varying degrees. Additionally, with time, some PTMs (especially Cd and Zn) of the slag might finally contaminate the groundwater by leaching and infiltration through its bottom weathering profile in the clastic rock region.

Introduction

Northwestern Guizhou is a concentrated region of the indigenous zinc smelting actives in Guizhou Province, with a long smelting history of more than 300 years (Wei et al., 2020; Yang et al., 2009; Zhou et al., 2020). There is still an vital need for study on the contamination of potentially toxic metal(loid)s (PTMs: i.e., arsenic (As), cadmium (Cd), chromium (Cr), copper (Cu), mercury (Hg), lead (Pb) and zinc (Zn)) in this region because they may be contaminating the surrounding environment and even endanger the health of animals or human beings due to PTMs of indigenous zinc slag. Randomly stacked and untreated slag has resulted in severe environmental pollution and risks in the indigenous zinc smelting region (Peng et al., 2018a). Although indigenous zinc smelting active has been banned by the local government for more than ten years (Peng et al., 2018b), there cannot be neglected the PTM contamination of the slag to their surrounding environment. The smelting slag-contaminated soils could increase the PTMs access by crops and vegetables (Liu et al., 2018), and the smelting activities are the dominant contribution to the Pb contamination in the mean of tissues of the maize from the Pb–Zn smelting area in southwest China (Wei et al., 2020). PTMs originated from smelter-originated wastes that are transferred into the soil, and then that might be finally caused to urgent health risks of the exposure of local residents (Zhou et al., 2020).

Potentially toxic metal(loid) contamination exists in the surface soil and soil profiles around the smelters or slag heaps in indigenous zinc smelting areas. Some researchers have investigated the surface soil surrounding the smelters or slag heaps (Verner et al., 1996; Ullrich, Ramsey & Helios-Rybicka, 1999; Bi et al., 2006; Yang et al., 2009; Li et al., 2015; Wang et al., 2015). They concluded that some PTMs were exceeded their risk-based screening values for the soil contamination of agricultural land (Ministry of Ecology and Environment of China, State Administration for Market Regulation, 2018). Additionally, in indigenous zinc smelting areas, soil profiles exhibit pollution by PTMs (especially in the surface soil), and the depths to which these elements migrate downward are significantly different. The Cd and Zn contamination depths could reach 2 m surrounding a Pb and Zn smelter in the north of France (Sterckeman et al., 2000). The Cu, Pb and Zn contents of the soil profile were close to constant around a Pb–Zn smelter in Baoji City, Northwest China (Wang et al., 2015). With increasing soil depths, the decreasing trend of Cd in the whole soil profile was unclear, and soil Pb and Zn contents decreased abruptly and respectively reached regional background levels at depths of 30 and 50 cm in Hezhang County, China (Bi et al., 2006). The Pb, Zn, Cu, Cr, As, Cd and Hg contents decreased to their background levels at a depth of approximately 150 cm around a smelter (Liu et al., 2015). Besides, comparing with the surface soil at depths ranging from 0 to 10 cm, there are polluted and enriched by Pb, Zn and As of the subsoil (depths ranging from 40 to 50 cm) of some locations near the town of Bytom where is an area of Pb/Zn mining and smelting in Upper Silesia, Poland (Ullrich, Ramsey & Helios-Rybicka, 1999). So there are many reports on the PTMs distribution of soil profiles, and most of them only focus on the soil profiles around the slag piles in indigenous zinc smelting areas (Scokart, Meeus-Verdinne & De Borger, 1983; Ullrich, Ramsey & Helios-Rybicka, 1999; Bi et al., 2006; Wang et al., 2015; Liu et al., 2015). But few studies are currently focusing on PTMs distribution and migration in the bottom weathering profile of the slag.

Additionally, to better understand the PTMs pollution in the bottom weathering profile caused by indigenous zinc smelting slag, it is necessary to take the control profile. A diagnosis of element soil pollution requires its pedogeochemical background content (Sterckeman et al., 2000). The pedogeochemical background content generally means the corresponding horizons of noncontaminated and same type soils (Baize, 1994; Sterckeman et al., 2000). Under the condition of no contamination, the contents of elements in the soil and soil profile depend on the composition of the parent material and the degree of soil formation (Zhou et al., 2002). There are differences in parent materials formed by natural weathering with different bedrock (Yang et al., 2011; Peng, Han & Chen, 2014).

Therefore, to better understand their contamination, distribution and migration of the soil section, we investigated the PTMs in the two weathering profiles (slag-covered and slag-absent) in the clastic rock region (where is one of the main stacking areas of the indigenous zinc smelting slag). This could help us better understand their PTMs contamination and provide some reasonable suggestions for better treating contamination of PTMs in the bottom weathering profile of the slag in the clastic rock.

Materials and Methods

Soil profile sampling

In August 2016, we collected 22 samples (approximately 1 kg each sample) from two weathering profiles (slag-covered and slag-absent) in an indigenous zinc smelting area in Honghualing of Dongfeng Town in Weining County, northwestern Guizhou Province, China. The soil is sandy loam weathered from the clastic rock. And the two weathering profiles had the same type of bedrock-weathered soil and were close to each other, which indicates that the two weathering profiles were under the same climatic and environmental conditions. So, their physical and chemical properties are similar.

As shown in Table S1, there were collected from the slag-covered weathering profile that a total of 12 samples (one slag sample and 11 soil samples). A total of 10 soil samples from a slag-absent soil profile, which was well covered by natural vegetation and not covered by slag, approximately 100 m away from the slag-covered section, were collected. Except that two upper soil sampling thicknesses of two weathering profiles (slag-covered and slag-absent) were several 10 and 20 cm, other soil sampling thicknesses of two sections were respective 20 and 25 cm until the dark gray sandy soil containing black granulating coal in their bottom layers.

Soil and slag sample analyses

The soil and slag samples were halved by applying the quartering method after removing foreign substances. One-half of each sample was dried at 30 °C to a constant weight in a thermostatic air-blower-driven drying closet and then sieved with a 10-mesh nylon sieve. The pH of each dry sieved sample (10.00 g of sample mixed with 25 ml deionized water) was determined by a pen pH meter (SX620 type, Instrument Factory of Shanghai Sanxin, Shanghai, China) (Peng et al., 2018a, 2018b; Peng, Chen & Yang, 2017).

And then the contents of PTMs (i.e., As, Cd, Cr, Cu, Hg, Pb and Zn) and Al are determined by an ICP-AES (America, Varian VISTA) and an ICP-MS (America, Agilent 7700x) in an accredited laboratory (ALS Minerals–ALS Chemex (Guangzhou) Co. Ltd.) (Peng et al., 2018a, 2018b; Peng, Chen & Yang, 2017). After grinding to 200-mesh (0.074 mm), each soil and slag sample was digested in two methods (Zhang et al., 2020). Firstly, 0.25 g of each sample was digested by a concentrated acid mixture (a ratio of 1:2.5:2:2.5 for the HClO4:HNO3:HF:HCl) in an oven at ~190 °C for 48 h, cooled to room temperature, heated on a preheated hot plate (150 °C) under a fume hood to get rid of excess acid until crystalline solid was formed, and then diluted to a steady volume (12.5 mL) with 2% hydrochloric acid. Secondly, 0.5 g of each sample was dissolved in aqua regia method (a ratio of 1:3 for HNO3:HCl) in an oven at ~190 °C for 48 h, and then was placed and heated on a preheated hot plate (150 °C) under a fume hood until white fumes appeared and crystalline solid was formed. Afterward, the crystalline solid was dissolved and precisely adjusted to a stable volume (12.5 mL) with deionized water.

Data statistics and analyses

Pollution index

Pollution index (PI) is a useful tool to express the degree of harm that is the ratio of the measured PTM content in the soil to the standard value of the PTM as follows:

PI=CiC0

where Ci and C0 represent the PTM content of the soil sample and its standard value of the PTM, respectively. Although many researchers used their local geochemical background as the element standard values, we selected the risk-based screening values for the soil contamination of agricultural land (Ministry of Ecology and Environment of China, State Administration for Market Regulation, 2018) (Table S1) as the element standard values. Because the former is its average value of the research area and maybe cause more errors to estimate the pollution. And the latter is the limited value for the soil contamination of agricultural land and could judge its contamination.

Enrichment factor

Enrichment factor (EF) is the relative abundance of an element, and it is a useful tool to differentiate the source of a chemical element, i.e., whether the source is anthropogenic or natural (Zhang et al., 2014; Peng, Chen & Yang, 2017; Peng et al., 2018b). The EF is calculated as follows:

EF=(MeAl)soil(MeAl)crust

Here, (Me/Al)soil and (Me/Al)crust refer to the soil and mean crustal content (Taylor, 1964) (Table S1) ratios, respectively, of a PTM and Al.

Transfer coefficient

Transfer coefficient (TC) is a useful tool to understand the element changes in a soil profile during natural weathering and the degree of element migration or enrichment. It could be used to calculate the loss or increase of elements in rock and soil during weathering (Nesbitt, 1979; Nesbitt, Markovics & Price, 1980; Brimhall, Alpers & Cunningham, 1985; Brimhall & Dietrich, 1987; Brimhall et al., 1991, 1992; Brantley & White, 2009; Zhang et al., 2018). The TC is calculated as follows:

TC=CipCiwCjwCjp−1

Here, the TC < 0 and TC > 0 is respective the fraction of a mobile element or mineral j lost or gained assuming that element or mineral i is immobile. C is the parent and weathered material element concentration (w and p present to the weathered and parent material respectively). The soil element content could be normalized by the content of an assumed immobile element Ti (Anderson, Dietrich & Brimhall, 2002; Brantley & White, 2009; Hausrath, Neaman & Brantley, 2009), Zr (Tian & Liu, 1995; Anderson, Dietrich & Brimhall, 2002; Brantley & White, 2009; Hausrath, Neaman & Brantley, 2009), Nb (Anderson, Dietrich & Brimhall, 2002; Brantley & White, 2009), Fe (Hausrath, Neaman & Brantley, 2009), Sc (Zhang et al., 2019) or Al (Tian & Liu, 1995; Hausrath, Neaman & Brantley, 2009) to calculate the fractional mineral loss or enrichment (Anderson, Dietrich & Brimhall, 2002; Brantley & White, 2009). In the chemical weathering process, Al is highly stable, and it is a good reference by which to measure the geochemical behavior of other elements (Chen et al., 1997). Therefore, Al is assumed to be the immobile element in this study.

Results

PTM concentrations and soil pH value in the weathering profiles (slag-covered and slag-absent)

The soil Cd, Pb, Zn concentrations might be mainly affected by these of the covered slag, other soil PTM contents and soil pH values are little or no affected by the covered slag. As shown in Table S1, the concentrations of soil As, Cd, Cr, Cu, Hg, Pb and Zn in the slag-covered section respectively range from 0.5 to 14.8 (mean 4.45), 0.61 to 4.59 (1.79), 59 to 77 (68.09), 102.0 to 164.0 (146.36), 0.025 to 0.090 (0.058), 7.7 to 191.5 (41.82) and 127 to 367 (231.36) mg/kg, and its pH values ranges from 4.42 to 5.22 (4.82). And their concentrations in the slag-absent section respectively range from 0.2 to 7.8 (mean 4.95), 0.14 to 1.82 (0.45), 49 to 94 (73.40), 84.5 to 135.0 (125.35), 0.027 to 0.110 (0.068), 5.7 to 19.6 (12.97), 61 to 202 (137.70) mg/kg, and its pH values ranges from 4.46 to 5.12 (4.94). These show that the soil Cd, Pb, Zn concentrations of the slag-covered section are notable higher than these of the slag-absent section, and the As, Cr, Cu and Hg concentrations and pH values of the soil in the slag-covered section are smilar to these of the slag-absent section. Additionally, the concentrations of As, Cd, Cr, Cu, Hg, Pb, Zn of the slag are respective 636, 192.0, 68, 499, 0.098, 10,550.0 and 16,450 mg/kg, and its pH value is 6.83. The slag As, Cd, Cu, Hg, Pb, Zn concentrations are more than these in the soil, and its pH value is very higher than the soil pH values of its bottom weathering profile. So, the slag Cd, Pb, Zn concentrations might impact these of its bottom weathering profile, and other PTM contents and pH value of the slag might be a little or no impact to these of its bottom weathering profile.

PTM distribution in the weathering profiles (slag-covered and slag-absent)

The soil As and Pb respective within 30 and 50 cm depth, and all soil Cd and Zn in the slag-covered weathering profile are remarkably contaminated by the indigenous zinc smelting slag. And the soil Cr, Cu and Hg of the slag-covered weathering profile might be little or no impacted by their covered slag. As shown in Fig. 1, the soil As and Pb contents respective within 30 and 50 cm depth, and all soil Cd and Zn concentrations in the slag-covered weathering profile are notably greater than those in the soil of the slag-absent section. These show that soil As and Pb respective within 30 and 50 cm depth, all soil Cd and Zn in the slag-covered weathering core, are remarkably polluted by the covered slag. However, there is little difference in the contents and change trends of Cr, Cu and Hg in the two weathering profiles (slag-covered and slag-absent). It shows that Cr, Cu and Hg in the whole slag-covered weathering profile might be little or no impacted by the covered slag.

Figure 1 Potentially toxic metal(loid) distribution in both slag-covered and slag-absent weathering profile in the study area.

(A–G) respectively represent the content distribution of As, Cd, Cr, Cu, Hg, Pb and Zn in the slag-covered weathering profile. (H–N) represent the concentration distribution of As, Cd, Cr, Cu, Hg, Pb and Zn in the slag-absent weathering profile, respectively.

Moreover, the soil As, Cd, Pb and Zn distribution of the slag-covered profile show they have an abrupt change with increasing soil depths, and they are different abrupt depths with different PTMs (Fig. 1). These abrupt changes might be caused by the As, Cd, Pb and Zn downward in the slag. There are different PTM mobility of the soil in different soil depths due to the soil physical and chemical properties (pH, texture, etc. (Ettler et al., 2007)). It might be the reason why these PTMs have different abrupt depths in this research area.

Additionally, the soil Hg distribution of the two weathering profiles is similar to the Liu, Li & Pan (2006) result. It shows a convex or irregular distribution (enrichment in the surface, middle and bottom layers to a certain extent). These explain that the soil Hg is polluted by industrial actives. Also, this similarity might be caused by Hg contamination from the soot produced during the smelting process is more seriously than Hg contamination from the slag (Protano & Nannoni, 2018). Because Hg in the original ore mainly evaporates first and diffuses into the surrounding environment through the indigenous zinc smelting soot due to the low melting point and boiling point of Hg. Therefore, the soil Hg concentration in the two weathering profiles is mainly contaminated by the indigenous zinc smelting soot but not the slag.

PTM migration in the weathering profiles (slag-covered and slag-absent)

The absolute content change of elements could not reflect their authentical geochemical behavior in the weathering and soil formation process. The mobile element leaching could make the immobile element enrich (Chen et al., 1997). It explains that the immobile element content might not reflect the authentical phenomenon of element migration or enrichment during the chemical weathering process. The TC > 0, TC < 0 and TC = 0 indicate that the element is enriched, depleted and immobile relative to the parent, respectively (Brantley & White, 2009; Hausrath et al., 2011). Also, the PTM proportion of the parent soil materials is on the foundation of their TC calculation of soil profiles. So there should be regarded theoretically as the effect of the element migration and redistribution in the soil geochemical process during the soil formatting and weathering process of the soil cores that any circumstance beyond the range of the TC value deviating from the total error due to analysis and determination (Tian & Liu, 1995). Therefore, when TC ≈ 0, the proportion of the element to its reference element in the soil maintains the parent material characteristics. And there is no notable element migration during the process of soil formation. When TC is significantly greater than 0, it indicates that the element is notably enriched relative to the parent. When TC is significantly lower than 0, it means that the element is remarkably depleted relative to the parent.

According to field observations, there is spring water near the weathering profile. And its elevation slightly lower than at the bottom layer elevation of the soil profile. These indicate that the groundwater level is relatively shallow (approximately the dept at 240 cm) in this research area. So, their bottom layer might be easily affected by the rising effect of groundwater capillary water. Additionally, the soil texture and color of the two bottom layers of the two weathering profiles are significantly different from their upper layers (Table S1). The two bottom layers of the soil profiles (slag-covered and slag-absent) are dark gray soil. These might be caused by the lower coal layer becoming soaked with upward capillary movement from a shallow water table (Huang & Xiu, 2010; Weil & Brady, 2016). Therefore, the penultimate layer might be a little affected by the bottom layer. And we chose the reciprocal third layer in the slag-absent profile as the reference of the parent layer in this study. The transfer coefficients of the PTMs in the two weathering profiles are presented in Fig. 2.

Figure 2 Transfer coefficients of potentially toxic metal(loid)s in both slag-covered and slag-absent weathering profile in the study area.

(A), (C), (E), (G), (I) and (K) respectively represent the transfer coefficient of As, Cd, Cr and Cu, Hg, Pb and Zn in the slag-covered weathering profile. And (B), (D), (F), (H), (J) and (L) respectively represent the transfer coefficient of As, Cd, Cr and Cu, Hg, Pb and Zn in the slag-absent weathering profile.

The soil As distribution is notably enriched within 110 cm depth in the slag-covered weathering profile. And its enrichment within 30 cm depth is remarkably higher than that in the slag-absent section. Additionally, the soil As content at soil depths between 50 and 190 cm in the slag-covered core is less than that in the slag-absent profile. The soil As is prominently depleted at soil depths ranged from 110 to 190 cm in the slag-covered core but not in the slag-absent section. According to yield observations, some plants are growing in the lower part of the slag-covered profile but not in the slag-absent section. Therefore, the soil As distribution difference might be related to the redistribution and biological absorption of As in the soil profile (Brantley & White, 2009).

The soil Cd is significantly enriched to varying enrichment degrees in the slag-covered profile, and the Cd content and enrichment degree are higher than those of the slag-absent core.

The soil Cr is completely depleted except for some enrichment in the bottom layer of the slag-covered core. It is no significant migration phenomenon in the slag-absent profile except for the soil layer at depths of 30–55 cm and 55–80 cm, and in the bottom layer.

The soil Cu shows no notable migration of the whole slag-covered profile. And it also shows no significant migration in the slag-absent section, except for a slight depletion at a depth of 30–130 cm and a light enrichment in the bottom layer.

The soil Hg is notably enriched within 110 cm depth and the bottom soil layer of the slag-covered core, and its enrichment degree is not higher than that in the slag-absent profile.

The soil Pb is remarkably enriched throughout the two soil profiles, and the Pb enrichment degree within 50 cm depth in the slag-covered core is higher than that in the slag-absent section.

The soil Zn is prominently enriched in the whole slag-covered profile, and its enrichment degree is higher than those in the slag-absent core.

These results show that the soil As and Pb respective within at least 30 and 50 cm depth, all soil Cd and Zn in the slag-covered core are affected by the downward migrations of those elements from the slag. However, the soil Cr, Cu and Hg of the slag-covered section might not be affected by the slag. Therefore, the slag PTM migration depths are different in the slag-covered weathering profile (i.e., the soil As and Pb correspond to within at least 30 and 50 cm depth, all soil Cd and Zn in the slag-covered core).

Discussion

PTM contamination in the soil profile impacted by the slag

The PI was introduced to evaluate and better understand the PTM contamination of soil profiles in indigenous zinc smelting areas in this research. A PI value greater than 1 indicates that the soil sample is polluted, while a PI value less than 1 suggests that the soil sample is unpolluted (Li et al., 2006). The 1 < PI < 3, 3 < PI < 5 and PI > 5 suggest that the soil is slightly polluted, moderately polluted and seriously polluted, respectively (Wu et al., 2015).

As shown in Table 1, the PIs of the slag As, Cd, Cu, Pb and Zn are greater than 5. It indicates that these PTM contents of the slag reach seriously polluted levels. If crops are planted on the slag, these PTMs are at risk of causing obvious pollution. The PIs of the slag Cr and Hg are less than 1, which indicates that their contents are required for their risk-based screening values for the soil contamination of agricultural land and that their contents are unpolluted. In the slag-covered soil profile, the PIs of soil As, Cr and Hg are less than 1, which indicates that their contents do not exceed the screening values of environmental pollution risk for agricultural land and do not reach the pollution level. The PIs of soil Cd and Cu are distributed in the ranges of 1–3, 3–5 and greater than 5, indicating that these PTMs are slightly, moderately and moderately polluted, respectively. The PIs of soil Pb within 30 cm depth and Zn in most soil samples range from 1 to 3, which indicate that their contents exceed the screening value of pollution risk in soil environmental quality and reach a slight contamination level. In the slag-absent soil profile, the PIs of soil As, Cr, Hg and Pb in the whole core and Cd and Zn in most soil samples are all less than 1. These mean that their contents do not exceed the screening value of pollution risk in soil environmental quality and do not reach a pollution level. There indicate that their contents are at a slight contamination level that the PIs of Cu in the whole core, Cd in soil within 30 cm depth and Zn in surface soil range from 1 to 3. Therefore, the soil Cd and Cu in the whole profile, the soil Pb within 30 cm depth, and Zn in most soil samples are contaminated to a certain extent in the slag-covered weathering profile. And the soil Cd within 30 cm depth, the Zn of surface soil, and all soil Cu in the slag-absent weathering core are slightly polluted.

Table 1 Pollution index of potentially toxic metal(loid)s in both slag-covered and slag-absent weathering profile in the study area.

Soil profile	Sample number and sampling depth (cm)	As	Cd	Cr	Cu	Hg	Pb	Zn	
Slag-covered	Slag	15.90	640.00	0.45	9.98	0.08	150.71	82.25	
S1 (0~10)	0.37	15.30	0.47	3.27	0.07	2.74	1.84	
S2 (10~30)	0.24	4.10	0.43	3.28	0.06	1.12	1.27	
S3 (30~50)	0.15	6.47	0.49	3.13	0.04	0.66	1.35	
S4 (50~70)	0.07	2.50	0.39	2.94	0.04	0.18	1.06	
S5 (70~90)	0.09	2.50	0.44	3.13	0.07	0.22	0.93	
S6 (90~110)	0.13	3.30	0.45	2.91	0.06	0.64	1.00	
S7 (110~130)	0.03	7.20	0.41	3.18	0.03	0.15	1.71	
S8 (130~150)	0.04	9.63	0.47	2.68	0.02	0.25	1.25	
S9 (150~170)	0.01	4.00	0.44	2.67	0.02	0.11	0.77	
S10 (170~190)	0.04	2.03	0.50	2.97	0.02	0.26	0.93	
S11 (190~210)	0.07	8.47	0.51	2.04	0.06	0.26	0.64	
Slag-absent	S1 (0~10)	0.17	6.07	0.52	2.64	0.08	0.28	1.01	
S2 (10~30)	0.17	2.13	0.48	2.55	0.08	0.21	0.75	
S3 (30~55)	0.17	0.63	0.49	2.66	0.08	0.21	0.72	
S4 (55~80)	0.20	0.57	0.63	2.70	0.05	0.23	0.72	
S5 (80~105)	0.19	0.50	0.51	2.68	0.04	0.22	0.71	
S6 (105~130)	0.14	0.47	0.51	2.58	0.05	0.22	0.70	
S7 (130~155)	0.09	0.57	0.51	2.61	0.03	0.12	0.65	
S8 (155~180)	0.05	0.67	0.48	2.60	0.02	0.08	0.68	
S9 (180~205)	0.07	0.70	0.45	2.36	0.04	0.13	0.67	
S10 (205~225)	0.01	2.77	0.33	1.69	0.04	0.15	0.31	

PTM source in the soil profile impacted by the slag

According to Sutherland (2000) and Chen et al. (2007), EF < 1, 1 ≤ EF < 3 , 3 ≤ EF < 5, 5 ≤ EF < 10, 10 ≤ EF < 25, 25 ≤ EF < 50 and EF ≥ 50 correspond to no enrichment, minor enrichment, moderate enrichment, moderately severe enrichment, severe enrichment, very severe enrichment, and extremely severe enrichment, respectively. The EFs of the PTMs could help further trace their sources in the soil (Hu et al., 2013; Peng, Chen & Yang, 2017). Although the element contents are mainly affected by parent rock types (Yang et al., 2010; 2011; Peng, Chen & Yang, 2017), elemental anomalies are mainly affected by specific natural geographical backgrounds (Yu et al., 2014) and by the effects of anthropogenic activities (Yu et al., 2014; Peng, Chen & Yang, 2017). The soil PTM contaminations might also be easily affected by anthropogenic activities, especially in surface soil (Peng, Chen & Yang, 2017). Moreover, an EF of less than one or greater than three indicates that the element originated predominantly from natural sources and anthropogenic activities, respectively (Hu et al., 2013). An EF value between 1 and 3 shows that the element is affected by both natural sources and anthropogenic activities.

As shown in Table 2, in the slag-covered weathering profile, the EFs of the soil As within 30 cm depth, the soil Pb and Zn within 50 cm depth, Cd in most soil samples are greater than 3; the EFs of As, Cd and Pb in a few soil samples, all soil Cu and Zn in most soil samples range from 1 to 3; other EFs are less than 1. In the slag-absent soil core, the EFs of the soil As within 105 cm depth and the soil Cd in the surface and bottom layers are greater than 3, the EFs of As, Cd, Hg and Pb in a few soil samples and Cu and Zn in the whole section range from 1 to 3, and other EFs are less than 1. These results show that As, Cd, Pb and Zn are mainly from anthropogenic activities contamination (i.e., the indigenous zinc smelting slag) at different depths of the slag-covered profile in the clastic rock district. Additionally, the As, Cd and Pb in a few soil samples, the Cu in the whole core, and the Zn in most soil samples of the slag-covered profile are mainly affected by human activities and natural weathering sources. Moreover, the soil Cd, Cr, Cu, Hg and Pb contents of the bottom layer are higher than those in the reciprocal third soil layer of the two weathering profiles. And the soil texture and color of the two bottom layers are significantly different from the upper soil in the weathering profiles. These might be caused by the upward capillary movement contaminations of the coal seam (below the bottom layer of the two weathering profiles). Therefore, the contents of PTMs (i.e., Cd, Cu, Pb and Zn) in the slag-covered profile are greatly affected by human activities (especially by the slag), while the Cd, Cu, Hg, Pb and Zn contents of the upper part in the slag-absent core might be affected by smelting activities.

Table 2 Enrichment factors of potentially toxic metal(loid)s in both slag-covered and slag-absent weathering profile in the study area.

Soil profile	Sample number and sampling depth (cm)	As	Cd	Cr	Cu	Hg	Pb	Zn	
Slag-covered	Slag	537.51	1460.41	1.03	13.80	1.86	1283.94	357.50	
S1 (0~10)	6.57	18.34	0.56	2.38	0.87	12.24	4.19	
S2 (10~30)	4.29	4.89	0.51	2.37	0.81	4.97	2.89	
S3 (30~50)	2.66	7.87	0.59	2.31	0.53	3.00	3.13	
S4 (50~70)	1.18	2.95	0.46	2.10	0.54	0.81	2.39	
S5 (70~90)	1.48	2.94	0.52	2.23	0.88	0.97	2.07	
S6 (90~110)	2.29	4.09	0.55	2.18	0.84	2.94	2.35	
S7 (110~130)	0.48	8.51	0.48	2.28	0.34	0.64	3.85	
S8 (130~150)	0.74	12.02	0.59	2.03	0.33	1.15	2.97	
S9 (150~170)	0.24	5.29	0.58	2.14	0.28	0.54	1.93	
S10 (170~190)	0.63	2.49	0.61	2.20	0.31	1.17	2.17	
S11 (190~210)	1.86	15.19	0.92	2.22	1.12	1.76	2.17	
Slag-absent	S1 (0~10)	3.11	7.72	0.66	2.04	1.17	1.33	2.45	
S2 (10~30)	3.13	2.69	0.61	1.95	1.10	0.99	1.79	
S3 (30~55)	3.11	0.77	0.59	1.96	1.07	0.97	1.66	
S4 (55~80)	3.45	0.68	0.75	1.95	0.69	1.03	1.64	
S5 (80~105)	3.45	0.61	0.61	1.97	0.58	1.01	1.64	
S6 (105~130)	2.47	0.56	0.61	1.89	0.69	0.98	1.60	
S7 (130~155)	1.73	0.72	0.64	2.00	0.46	0.57	1.55	
S8 (155~180)	0.92	0.92	0.66	2.18	0.31	0.42	1.78	
S9 (180~205)	1.37	1.00	0.64	2.03	0.60	0.69	1.80	
S10 (205~225)	0.18	6.70	0.79	2.48	0.93	1.33	1.41	

PTM migration and its influence in the soil profile impacted by the slag

Some PTMs might be released by natural weathering and entered into the surrounding environment by leaching rain. Some PTMs (i.e., As, Cd, Cu, Hg, Pb and Zn) of the slag are very high and more than these in the soil of its bottom weathering profile (Table S1). Although most of the slag PTMs are generally dominated by polymetallic or other phases (Ettler, Piantone & Touray, 2003; Scokart, Meeus-Verdinne & De Borger, 1983; Sobanska et al., 2016), they could be easily transported to the surrounding circumstances due to long-term natural weathering (Sobanska et al., 2016; Tyszka et al., 2014, 2018) or other processes occurring in acidic environments (Scokart, Meeus-Verdinne & De Borger, 1983; Sobanska et al., 2016; Warchulski et al., 2019; Yang et al., 2006). There adds to the bioavailability of Cd that a possible transformation of Cd from metal oxides in smelting slags to adsorbed phases and carbonates (Wang et al., 2020). Warchulski et al. (2019) prove the importance of rainfall-induced weathering on PTMs mobilization and migration.

Some PTMs of the slag might be migrated downward and contaminated the groundwater by leaching and infiltration through the soil profile in the clastic rock distribution district. The contents of some PTMs (i.e., Cd, Cu, Pb and Zn) in the slag-covered weathering profile are greatly affected by the covered slag in this research area. As described in Fig. 2, the soil As and Pb respective within at least 30 and 50 cm depth, all soil Cd and Zn in the slag-covered weathering profile are affected by the slag. And the groundwater level is relatively shallow (approximately the dept at 240 cm) in the research area through field observations. Additionally, many leaching experiments show that some PTMs in the slag could easily migrate into the surrounding environment (Zhu et al., 2012; Piatak, Parsons & Ii, 2015; Liu et al., 2018; Tyszka et al., 2018; Warchulski et al., 2019). The Cd and Zn migrate with organic matter that brings about more potential hazards of the groundwater movements near zinc smelters in acidic sandy soils (Scokart, Meeus-Verdinne & De Borger, 1983). Groundwater quality is mostly poor near the Pb–Zn mining and smelter and coal mining districts in Hezhang County where borders on Weining Prefecture in Guizhou Province (Zhao et al., 2016). The heavy Zn isotopic signatures in the groundwater might be concerned to seepage from the slag piles by the advective or diffusive transport of pore water from the polluted soil at the smelting site (Yin et al., 2016).

Therefore, over time, some slag PTMs (especially Cd and Zn) would be released by natural weathering and re-migrated into the soil of its bottom weathering profile by leaching rain, and eventually infiltrate into deeper soil, and even might contaminate the groundwater.

Conclusions

In the clastic rock region, the soil Cd and Cu in the whole profile, the soil Pb within 30 cm depth and Zn in most soil samples are to a certain extent at pollution levels in the slag-covered weathering profile. All soil Cu, the soil Cd within 30 cm depth and Zn in the surface soil in the slag-absent section are slightly contaminated. The contents of PTMs (i.e., Cd, Cu, Pb and Zn) are greatly affected by human activities (especially by the slag) in the slag-covered profile. And the contents of PTMs (i.e., Cd, Cu, Hg, Pb and Zn) in the upper part of the slag-absent core might be affected by the indigenous zinc smelting activities.

Additionally, the contents of the soil As and Pb respective within 30 and 50 cm depth, all soil Cd and Zn in the slag-covered weathering core are higher than those in the slag-absent profile. And the soil As is enriched within 110 cm depth (especially within 30 cm depth). All soil Cd, Pb and Zn are enriched to varying degrees in the whole slag-covered profile (especially the soil Pb within 50 cm depth).

Moreover, the soil As, Cd, Pb and Zn of the slag-covered soil profile are affected by the slag in the clastic rock region, and their affected depths and degrees are different. The soil As and Pb respective within 30 and 50 cm depth, and all soil Cd and Zn in the slag-covered profile are remarkably affected by the downward migration of these PTMs from the slag. All soil Cr, Cu and Hg of the slag-covered section are not or are little affected by the slag. Additionally, with time, some PTMs (especially Cd and Zn) of the slag might contaminate the groundwater by leaching and infiltration through the soil profile in the clastic rock region. Therefore, there should be an urgency to manage untreated, scattered stacking slag to further prevent the contamination of the soil and the groundwater in the research area by PTMs.

Supplemental Information

Supplemental Information 1 Soil characteristics and potentially toxic metal(loid) concentrations in both slag-covered and slag-absent weathering profile in the study area.

Click here for additional data file.

Supplemental Information 2 Raw data.

Click here for additional data file.

We are very grateful to the anonymous reviewers for their helpful comments and constructive criticisms of both the English and the technical writing of this manuscript.

Additional Information and Declarations

Competing Interests

Author Contributions

Data Availability

The authors declare that they have no competing interests.

Yishu Peng performed the experiments, analyzed the data, prepared figures and/or tables, authored or reviewed drafts of the paper, and approved the final draft.

Ruidong Yang conceived and designed the experiments, authored or reviewed drafts of the paper, and approved the final draft.

Tao Jin analyzed the data, prepared figures and/or tables, and approved the final draft.

Jun Chen analyzed the data, prepared figures and/or tables, and approved the final draft.

Jian Zhang analyzed the data, prepared figures and/or tables, and approved the final draft.

The following information was supplied regarding data availability:

Raw data are available in the Supplemental Files.

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
