# Peer review of "Potentially toxic metal(loid) distribution and migration in the bottom weathering profile of indigenous zinc smelting slag pile in clastic rock region"

_PeerJ, doi:10.7717/peerj.10825_

## Round 0.1 · original submission · Major Revisions

Dear Author,

I am glad to be the Academic Editor of this interesting manuscript that determined concentrations of PTMS and pH values and analyzed PTM distribution in two weathering profiles with small horizontal distance....but you need to improve some aspects indicated for reviewers before acceptance.

In general terms you need to check and improve your grammar, you should add some more recent citations and remember to add a comment that states the contribution the article makes to the literature. Finally, I want to encourage you to send us the indicated improvement as soon as possible.

Reviewer 1 ·

Basic reporting

This manuscript prsents a study determined the concentrations of PTMs and pH values and analyzed PTM distribution in two weathering profiles (slag-covered and slag-absent) with small horizontal distance in clastic rock region in the indigenous zinc smelting area.
Generally speaking, the research content of this study is detailed, and it is very interesting.The manuscript is wirtten clearly and unambiguously. Obviously,the authors have done a lot of work in writing this manuscript.However, there are still some problems need to be considered before acceptance.

Experimental design

It is suggested that the author should refer to more recent references and explain clearly in introduction how this research fills the identified knowledge gap.
In part 2.1 of the paper, it is suggested to describe in more detail the sampling method and the principles adopted in the sampling process, as well as how to sample at the bottom weathering profile.

Validity of the findings

It is suggested that results and discussion should be written in a chapter, or a section on migration of potentially toxic metals (PTM) should be added to the discussion part separately.
It is suggested that more detailed supplementary support should be given to support the conclusion that “the downward migration of Cd and Zn from the covered slag might finally contaminate to groundwater through the bottom weathering profile in the clastic rock region”.

Additional comments

Line71 to line 72. The site location information for lead, zinc and arsenic contaminated subsoil within 40 to 50cm depth of soil is required to be added when comparing with the one in Poland.
Please check the details of the article carefully and pay attention to the writing and drawing specifications of the paper. For example,the unit of flagis suggested to be added in Figure 1.The unit of flagis suggested to be added in Figure 1.
Therefore,the manuscript may be reviewed after major modification.

·

Basic reporting

I have completed the review of the manuscript “Potentially toxic metal(loid) distribution and migration in the bottom weathering profile of indigenous zinc smelting slag pile in clastic rock region” by Peng et al. This is a very interesting manuscript that provides pedogeochemical data of two weathering profiles (slag-covered and slag-absent) in an indigenous zinc smelting slag site located in Guizhou Province, China. However, the discussion section must be significantly improved before publication. In particular, the relevance of the clastic region in the potential dispersion or stability of potentially toxic metal(loid)s is not discussed.

General
I strongly advise to use professional English editing services to improve the grammar in your manuscript. Examples are lines 40-41, 54-55,


The introduction section needs significant improvement. I suggest to include a brief description of indigenous metal smelting slag and its environmental significance.

Table 1. It is unclear what do the authors mean with “a minimum of soil”, “maximum of soil”, “mean of soil”.

Figure 1 repeats information from Table 1. I suggest to include the table in a Supplementary Material file.

Line 48: Peng et al., 2018b, it should be 2018a.
Please provide a description of the study area located at the beginning of the Introduction section. Lines 47 to 51 are unclear to the unfamiliar reader.

Experimental design

Lines 116-118. Please provide the sample digestion method, as well as detection limits for all studied metal(loid)s.

Line 124. The pollution index does not specifically determine the degree of harm. Please re-phrase. To accurately estimate the Pollution Index (PI), authors should provide a discussion of the local geochemical background, if such background is not used in the calculation of PI, then using risk-based screening values for the soil contamination of agricultural land should be supported by a discussion.

Validity of the findings

The discussion section needs improvement. The impact and novelty of this research is not assessed.

Additional comments

I have completed the review of the manuscript “Potentially toxic metal(loid) distribution and migration in the bottom weathering profile of indigenous zinc smelting slag pile in clastic rock region” by Peng et al. This is a very interesting manuscript that provides pedogeochemical data of two weathering profiles (slag-covered and slag-absent) in an indigenous zinc smelting slag site located in Guizhou Province, China. However, the discussion section must be significantly improved before publication. In particular, the relevance of the clastic region in the potential dispersion or stability of potentially toxic metal(loid)s is not discussed.

General
I strongly advise to use professional English editing services to improve the grammar in your manuscript. Examples are lines 40-41, 54-55,

Lines 116-118. Please provide the sample digestion method, as well as detection limits for all studied metal(loid)s.

Line 124. The pollution index does not specifically determine the degree of harm. Please re-phrase. To accurately estimate the Pollution Index (PI), authors should provide a discussion of the local geochemical background, if such background is not used in the calculation of PI, then using risk-based screening values for the soil contamination of agricultural land should be supported by a discussion.

The introduction section needs significant improvement. I suggest to include a brief description of indigenous metal smelting slag and its environmental significance.

Table 1. It is unclear what do the authors mean with “a minimum of soil”, “maximum of soil”, “mean of soil”.

Figure 1 repeats information from Table 1. I suggest to include the table in a Supplementary Material file.

Line 48: Peng et al., 2018b, it should be 2018a.
Please provide a description of the study area located at the beginning of the Introduction section. Lines 47 to 51 are unclear to the unfamiliar reader.

---

## Round 0.2 · accepted · Accept

Dear author, I have reviewed that you have complied with the comments of the reviewers, which has improved your manuscript. I would like to congratulate you along with the co-authors for important contribution on the subject of Potentially toxic metal (loid) distribution and migration in the bottom weathering profile of indigenous zinc smelting slag pile in clastic rock region.